# Circulating Tumor DNA Profiling of a Diffuse Large B Cell Lymphoma Patient with Secondary Acute Myeloid Leukemia

**DOI:** 10.3390/cancers14061371

**Published:** 2022-03-08

**Authors:** Irina A. Kerle, Ludwig Jägerhuber, Ramona Secci, Nicole Pfarr, Philipp Blüm, Romina Roesch, Katharina S. Götze, Wilko Weichert, Florian Bassermann, Jürgen Ruland, Christof Winter

**Affiliations:** 1Department of Medicine III, Klinikum Rechts der Isar, School of Medicine, Technical University of Munich, 81675 Munich, Germany; irina.kerle@nct-dresden.de (I.A.K.); philipp.bluem@mri.tum.de (P.B.); katharina.goetze@tum.de (K.S.G.); florian.bassermann@tum.de (F.B.); 2Center for Personalized Oncology, National Center for Tumor Diseases (NCT) Dresden and University Hospital Carl Gustav Carus Dresden at TU Dresden, 01307 Dresden, Germany; 3Institute of Clinical Chemistry and Pathobiochemistry, Klinikum Rechts der Isar, School of Medicine, Technical University of Munich, 81675 Munich, Germany; l.jaegerhuber@googlemail.com (L.J.); ramona.secci@tum.de (R.S.); romina.roesch@tum.de (R.R.); j.ruland@tum.de (J.R.); 4Institute of Pathology, School of Medicine, Technical University of Munich, 81675 Munich, Germany; nicole.pfarr@tum.de (N.P.); wilko.weichert@tum.de (W.W.); 5German Cancer Consortium (DKTK), Partner Site Munich, 81675 Munich, Germany; 6German Cancer Research Center (DKFZ), 69120 Heidelberg, Germany; 7TranslaTUM, Center for Translational Cancer Research, Technical University of Munich, 81675 Munich, Germany

**Keywords:** liquid biopsy, cell-free DNA, digital PCR, precision medicine

## Abstract

**Simple Summary:**

Liquid biopsy is a diagnostic procedure in which a blood sample taken from a cancer patient is searched for cell-free DNA that originates from tumor cells in the body. Analysis and quantification of this so-called circulating tumor DNA (ctDNA) can help to monitor therapy response and help to detect relapse earlier than routine clinical diagnostics. In this study, we report on a patient diagnosed with lymphoma who developed therapy-related acute leukemia and whom we profiled with liquid biopsy in blood samples taken at 13 time points over 26 weeks. Our liquid biopsy strategy used a combination of next-generation sequencing (NGS) and tumor-specific digital polymerase chain reaction (dPCR) assays. This strategy revealed not only lymphoma mutation dynamics during treatment, but was also able to capture the onset of therapy-related acute leukemia by detecting two leukemia specific mutations. We conclude that liquid biopsy based on analyzing ctDNA by combining targeted NGS with mutation-specific dPCR is a feasible tool for monitoring treatment response in lymphoma patients and is also capable of detecting therapy-related acute leukemia.

**Abstract:**

Diffuse large B cell lymphomas (DLBCL) are the most common neoplasia of the lymphatic system. Circulating cell-free DNA released from tumor cells (ctDNA) has been studied in many tumor entities and successfully used to monitor treatment and follow up. Studies of ctDNA in DLBCL so far have mainly focused on tracking mutations in peripheral blood initially detected by next-generation sequencing (NGS) of tumor tissue from one lymphoma manifestation site. This approach, however, cannot capture the mutational heterogeneity of different tumor sites in its entirety. In this case report, we present repetitive targeted next-generation sequencing combined with digital PCR out of peripheral blood of a patient with DLBCL relapse. By combining both detection methods, we were able to detect a new dominant clone of ctDNA correlating with the development of secondary therapy-related acute myeloid leukemia (t-AML) during the course of observation. Conclusively, our case report reinforces the diagnostic importance of ctDNA in DLBCL as well as the importance of repeated ctDNA sequencing combined with focused digital PCR assays to display the dynamic mutational landscape during the clinical course.

## 1. Introduction

Diffuse large B cell lymphomas (DLBCL) are the most common neoplasia of the lymphatic system and present as aggressive malignancies with often acute need for therapy [1,2]. In Europe, the incidence rate is about 4 cases per 100,000 persons per year [3]. Physical and blood examination, tumor tissue biopsy, and clinical imaging such as computed tomography (CT) or positron emission-CT (PET-CT) are the main pillars of the diagnostic procedure [4].

The standard immunochemotherapy is R-CHOP, which contains the CD20 antibody rituximab as well as the chemotherapeutic agents cyclophosphamide, doxorubicin, vincristine and the glucocorticoid prednisone [5]. Chemotherapy, however, increases the risk of secondary malignancies such as therapy-related acute myeloid leukemia (t-AML) [6]. Retrospective studies found a standardized incidence ratio for t-AML after immunochemotherapy for DLBCL between 5.0 and 8.7 [7,8]. Treatment of t-AML can range from curative approaches with high-dose anthracycline and cytarabine protocols followed by allogeneic stem cell transplantation, to palliative protocols with azacitidine or low-dose decitabine/cytarabine.

In recent years, the analysis of cell-free DNA (cfDNA) and circulating tumor DNA (ctDNA), known as Liquid Biopsy (LB), has emerged in order to improve treatment monitoring and follow up of solid and hematologic malignancies [9,10,11,12]. Several studies showed that ctDNA encoding tumor-specific immunoglobulin rearrangements in patients with DLBCL is a representative tumor marker associated with tumor stage, prognosis and risk factors [13,14,15].

Many LB studies in DLBCL have employed next generation sequencing (NGS) of tumor tissue from one lymphoma manifestation site (mostly from a lymph node biopsy at first diagnosis) followed by tracking selected mutations in the peripheral blood by polymerase chain reaction (PCR). Such techniques rely on mutations present at diagnosis; hence, they cannot capture tumor evolution, and bear the risk of detecting only a subset of the mutational profile due to the possible tumor heterogeneity of different manifestation sites. 

An approach to resolve this issue is the employment of NGS of cfDNA in peripheral blood, which can be viewed as a collection of repetitive biopsies of different tumor sites. Here, we report on cfDNA monitoring by NGS complemented by digital PCR on serial peripheral blood samples of a patient diagnosed with DLBCL relapse followed by secondary t-AML. 

## 2. Materials and Methods

### 2.1. Patient and Samples

The patient took part in a prospective non-interventional liquid biopsy study (LYMPHCIN) for non-Hodgkin B cell lymphomas carried out at the University Hospital of the Technical University of Munich, Germany (2017–2020). Peripheral venous blood samples for the study were taken at first diagnosis/diagnosis of relapse and at each following visit to the hospital due to treatment, staging appointments or follow up. Blood plasma was obtained by double centrifugation of EDTA blood tubes and then stored at −80 °C. 

### 2.2. Targeted NGS of cfDNA and Tumor-Specific ddPCR Assays

We used the Avenio Expanded Kit (Roche Diagnostics, Mannheim, Germany) comprising 77 genes for targeted NGS of cfDNA (Appendix A). To this end, cfDNA was isolated from stored plasma using the QIAamp Circulating Nucleic Acid Kit (Qiagen, Hilden, Germany) and quantified on a Qubit 2.0 fluorometer (Thermo Fisher Scientific, San Jose, CA, USA). Libraries were generated from 50 ng of plasma DNA and sequenced on a NextSeq 500 system (Illumina, San Diego, CA, USA). Sequencing data were analyzed by Roche Diagnostics, Mannheim, Germany. After detecting patient-specific somatic mutations in plasma by targeted NGS, tumor- and wild-type-specific droplet digital PCR (ddPCR) assays for ctDNA quantification were designed and validated by internal controls and by lymphoma tissue extracted at initial diagnosis. A QX200 ddPCR system with automated droplet generation (Bio-Rad Laboratories) was used and PCR was carried out on a C1000 Touch Thermal Cycler (Bio-Rad). Finally, plates were read on a QX200 droplet reader (Bio-Rad) to determine droplet fluorescence intensities.

### 2.3. Droplet Digital PCR Data Analysis

Raw droplet fluorescence intensity values were exported from QuantaSoft droplet reader software v1.7.4 (Bio-Rad). Custom scripts were used to import the intensity values into R (version 3.4.4) and to quantify concentrations of mutant and wild-type DNA. Target concentrations (c) were calculated for each well from the number of positive droplets (Np) and negative droplets (Nn) and the average droplet volume (V) = 0.85 nanoliters based on Poisson distribution statistics using the formula c = (ln(Np + Nn) − ln(Nn))/V, where ln is the natural logarithm. 

## 3. Results and Discussion

The study goal was to evaluate the potential of individual tumor-specific cfDNA markers detected by interim targeted sequencing of cfDNA in blood plasma samples for treatment and relapse monitoring of patients with non-Hodgkin B cell lymphoma. We herewith present one of our analyzed cases of a patient with relapsed DLBCL and subsequent development of secondary, therapy-associated acute myeloid leukemia.

### 3.1. Patient Course of Treatment

Study subject is a 72-year-old male first diagnosed with non-germinal center B-cell-like (non-GCB) type DLBCL Ann Arbor stage IVB with lymphoma manifestations around the spleen and para-aortic lymph nodes. The diagnosis of non-GCB type DLBCL was established by biopsy of para-aortic lymph node masses followed by histopathological morphological as well as immunohistochemical workup (Figure 1A, see Appendix A for details). Except for an elevated serum lactate dehydrogenase (LDH) level of 375 U/L, laboratory findings at diagnosis were unremarkable. The patient underwent six courses of R-CHOP-14 (rituximab 375 mg/m^2^, cyclophosphamide 750 mg/m^2^, doxorubicin 50 mg/m^2^, vincristine 1 mg i.v. day 1, prednisone 100 mg p.o. day 1–5, short acting G-CSF (filgrastim) s.c. day 4 until granulocyte regeneration; q14) followed by two subsequent administrations of rituximab monotherapy. PET-CT at the end of treatment showed a complete remission.

Six months after the end of treatment, recurrent disease was diagnosed in the shape of de novo perisplenic lymphoma manifestations detected by CT and PET-CT (Figure 2A,B). Due to an impaired general condition and a history of acute coronary syndrome episodes as a result of underlying coronary artery disease, a high-dosage chemotherapeutic approach followed by autologous stem cell transplantation was omitted. Instead, the patient was enrolled and treated within the chemotherapeutic-free PCYC1123-Ca trial (rituximab 375 mg/m^2^ i.v. day 1, lenalidomide 20 mg p.o. d1–21 q28, ibrutinib 560 mg p.o. daily). Blood sample collection for our study was initiated (Figure 3A) by start of treatment for relapse. Due to neutropenia CTC grade III, treatment cycles 3 and 4 were postponed for 7 days, respectively. In order to avoid further treatment delays, filgrastim was administered for four subsequent days before cycle 4, which led to sufficient granulocyte rise. PET-CT restaging after three cycles showed a partial remission (Figure 2C). Due to neutropenia CTC grade IV, cycle 5 was postponed again and filgrastim was readministered for four subsequent days. This was followed by a massive rise in leukocyte count (71 G/L, 76% myeloid blasts, Figure 3C), which led to the diagnosis of AML (subtype acute myelomonocytic leukemia), established by morphological and immunophenotypical characterization of peripheral blood (Figure 1B, see Appendix A for details).

Lymphoma treatment was subsequently terminated and antileukemic therapy was initiated. At this point in time, the patient’s condition deteriorated immensely; thus, no extensive diagnostic workup (bone marrow biopsy, standard molecular and cytogenetic examination of blast population) was conducted. The patient was treated with mild, palliative therapy regimens for AML (cytarabine 100 mg i.v. per day, followed by azacitidine 75 mg/m^2^ s.c. day 1–7). After four weeks with neither laboratory nor clinical therapy response, antileukemic therapy was terminated and switched to best supportive care. Shortly after, the patient died as a result of therapy refractory secondary hematologic malignancy. 

### 3.2. Mutational Profiling of Cell-Free Tumor DNA in Blood Plasma

Blood samples representing 13 different time points were collected for cfDNA analysis, covering relapse diagnosis, treatment, and imaging appointments (see Figure 3).

In order to create a tumor ctDNA profile, we performed targeted sequencing on 4 of the 13 blood samples (marked with asterisks in Figure 3A). This analysis revealed a total of 106 mutations (Appendix A). Of these, 87 were germline, and 19 were somatic, the latter affecting 14 genes. We selected seven somatic mutations with a variant allele frequency of at least 0.25% for designing mutation-specific digital PCR assays, and used these to quantify ctDNA concentrations in all 13 plasma samples and in lymphoma tissue retrieved at initial diagnosis. 

Of note, only four out of seven mutations were present in lymphoma tissue, namely *TP53* p.R273L, *PDGFRB* p.T140M, *KRAS* p.G13D, and *PDGFRA* p.T276M. The first two mutations appeared to be patient individual driver mutations for DLBCL during the observed time period, since they had considerable plasma concentrations, and showed high mutual correlation and correlation with LDH levels (Figure 3A,B). 

*TP53* p.R273L is an oncogenic missense mutation according to Variant Interpretation for Cancer Consortium (VICC) criteria [16]. It has been described as gain of function or driver mutation in numerous malignancies [17,18,19]. *TP53* p.R273 mutations are known tumor hotspot mutations and have been detected in DLBCL recurrences, interpreted as evolutionary alterations of therapy resistance [15,20]. 

*PDGFRB* p.T140M is a missense variant which, according to our knowledge, has not yet been described in the literature. Since *PDGFRB* p.T140M changes in our patient closely mimic those of the known oncogenic *TP53* p.R273L, this mutation is likely part of the same subclone. Its oncogenic potential, however, is unclear, and it could well be a passenger mutation that is still of diagnostic value in this particular tumor.

After initiation of antilymphoma therapy, ctDNA levels showed a vast decline (nadir at 20 December 2017), indicating a treatment response four weeks before the first interim PET-CT (18 January 2018). The delay of the third therapy cycle triggered a subsequent rise of ctDNA, followed by a new fall through the application of cycle 4. These ctDNA dynamics reflect a highly volatile tumor biology not displayed by the interim PET-CT, emphasizing the predictive potential of ctDNA for treatment monitoring in lymphomas [21,22].

Two mutations, *FLT3* p.D839G and *FLT3* p.D835H (verified oncogenic mutations of the tyrosine kinase domain of FLT3 [23,24]), were first detected at time of diagnosis of AML and, thus, were not present in the initial tissue sample. Since *FLT3* p.D839G showed an exponential rise simultaneous to blast excess, it seems to represent the patient-specific driver mutation for AML. *FLT3* p.D839G is a gain of function mutation located within the activation loop of the tyrosine kinase domain (TKD) of the protein. It is both described as a baseline mutation [25] as well as a secondary mutation of AML at relapse [26,27]. Of utmost significance is the characterization of *FLT3* TKD mutations at residues D835, D839, and Y842 as frequent mechanisms of FLT3 inhibitor resistance, hence being therapy-related [28].

Owing to prolonged malignant illness in the shape of relapsed DLBCL, the presented patient was exposed to extensive cytotoxic chemotherapy treatment at first diagnosis and, again, received targeted and immunomodulatory therapy at relapse. The alkylating agent cyclophosphamide, the anthracycline doxorubicin as well as the vinca alkaloid vincristine were all previously linked to the risk of development of therapy-related MDS/AML [29,30,31,32].

A retrospective cohort study analyzing the role of granulocyte colony-stimulating factors (G-CSFs) in the development of MDS and AML in patients with chemotherapeutically treated non-Hodgkin lymphoma showed a higher incidence of MDS/AML in patients who received short acting G-CSF (filgrastim) at higher frequency (≥10 doses), especially among those patients who received multiple chemotherapy regimens [33]. Through these and other convincing data of breast cancer patients formerly treated with anthracyclines and/or cyclophosphamide [34], a contributing role of G-CSF in the development of t-AML in our patient’s case cannot be excluded, since short-acting filgrastim was administered in both treatment at first diagnosis as well as in relapse treatment. 

To this day, apart from anecdotal case reports, there is no significant indication that the administration of G-CSF itself leads to the development of myeloid malignancies in healthy subjects [35,36,37]. In synopsis with morphological, immunophenotypical and molecular results combined with the patient’s chemotherapeutic history, we therefore strongly believe that there is substantive evidence of therapy-related AML after recurrent cytotoxic treatment for DLBCL. The repeated administration of G-CSF could have promoted leukemic cell survival and, ultimately, triggered blast outwash into the peripheral blood stream; there is yet no evidence of a causal relationship. 

The adynamic, constant detection of the known oncogenic hotspot mutation *TP53* p.G245S [38,39] and *PDGFRA* p.T276M (Figure 3A) could either be interpreted as these alterations being non-dominant lymphoma-dependent mutations during the monitored time period or, on the other hand, they could represent tumor-independent mutations known as CHIP (clonal hematopoiesis of indeterminate potential). In order to further analyze this circumstance, plasma collection for ctDNA prior to DLBCL relapse is needed, which unfortunately is not available.

### 3.3. Limitations

Due to the consumption of the entire available sample plasma volume for NGS-based analysis of cfDNA, we could not directly compare ddPCR and NGS results for the same plasma sample. The NGS panel used in this study contained 77 genes commonly mutated in a variety of tumor entities; hence, not all oncogenes and tumor suppressor genes of interest were included and assayed. In addition, preparation and conduct of NGS for liquid biopsies are both time- and resource-intensive and, therefore, are often limited to study settings within university-associated treatment centers.

## 4. Conclusions

Our case report shows that the LB approach of combining interim targeted NGS with mutation-specific digital PCR is a feasible tool for monitoring treatment response in DLBCL. This combined monitoring approach allows for a more precise depiction of lymphoma activity and is capable of detecting genetic alterations associated with secondary malignancies such as t-AML. In order to maximize the capability of LB in further studies, targeted, preferably whole exome sequencing, at regular time points during treatment of DLBCL is warranted.

## Figures and Tables

**Figure 1 cancers-14-01371-f001:**
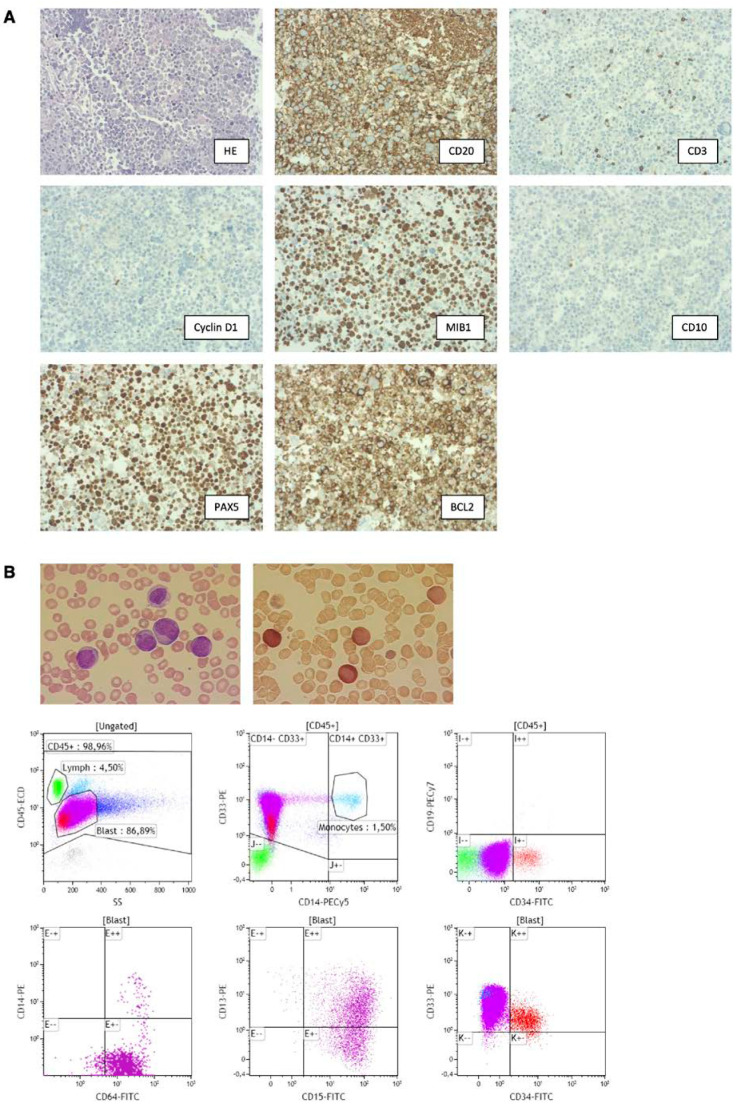
Diagnostic workup of DLBCL and AML of patient. (**A**) Fine needle biopsy of an abdominal lymph node showing an aggressive B-cell-lymphoma expressing CD20, PAX5 and BCL2, without expression of CD3 (some admixed non-neoplastic T-cells are positive), Cyclin D1, and CD10, corresponding to a diffuse large B-cell-lymphoma (DLBCL). MIB1 staining indicates expression of Ki-67, a nuclear protein associated with cell proliferation. (**B**) Diagnostic workup from the peripheral blood for AML. Upper panel: cytomorphology. Upper left: Pappenheim stain showing blasts with monocytic morphology and prominent nucleolus. Upper right: esterase staining showing esterase positive blasts. Lower panel: immunophenotyping showing AML blasts with positive expression of CD33+ CD64+ CD13+ and CD15+ and a subpopulation with CD34+, indicative of myelomonocytic leukemia. No expression of B cell marker CD19.

**Figure 2 cancers-14-01371-f002:**
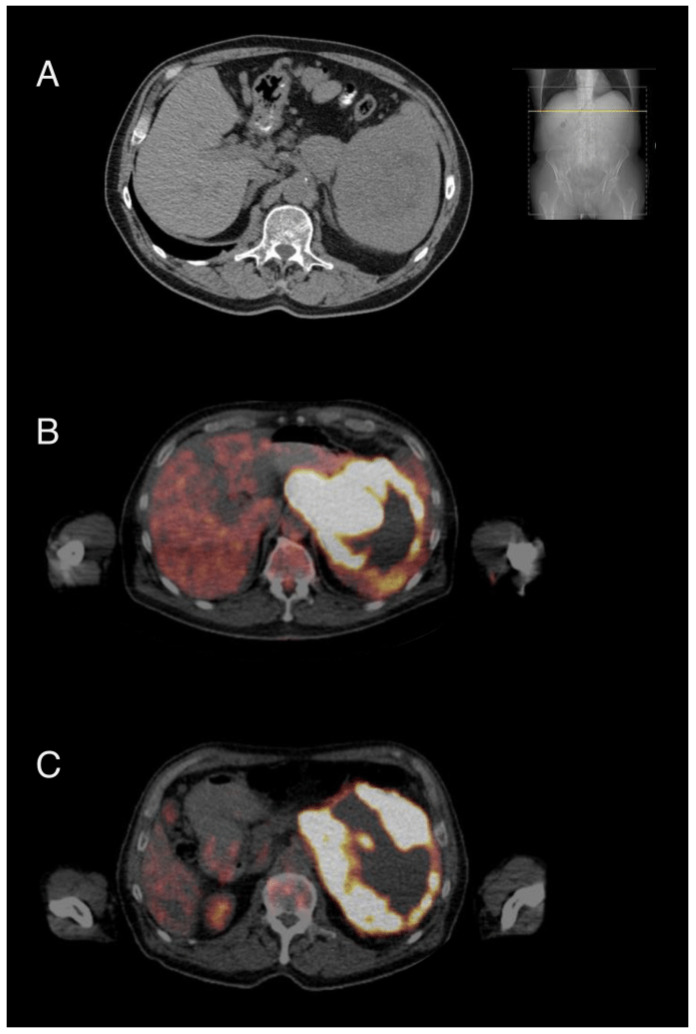
Clinical imaging of patient. (**A**) CT scan on 27 September 2017. Detection of DLBCL relapse with lymphoma manifestations around the splenic hilum. (**B**) PET-CT scan on 18 October 2017. Strong fluorodeoxyglucose metabolism in the area of the splenic hilum indicating active lymphatic tumor masses. (**C**) PET-CT scan on 18 January 2018. Regressive areas of glucose metabolism indicating regressive lymphatic tumor masses in the sense of a partial response.

**Figure 3 cancers-14-01371-f003:**
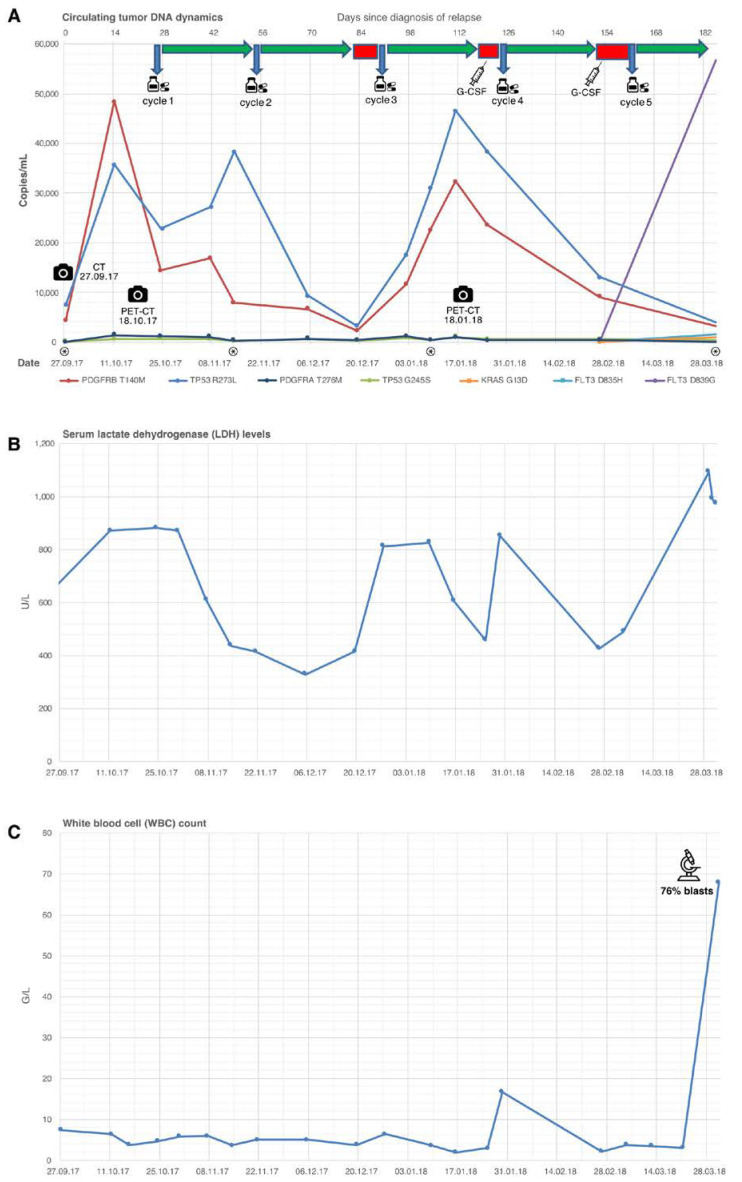
Time course of patient’s disease and treatment. (**A**) Amount of ctDNA in the patient’s blood in copies per milliliter. Above the diagram, the therapy regimen is shown. Blue vertical arrow/pill box, first day of each new treatment cycle; green arrow, treatment period; red box, therapy interruption due to neutropenia; syringe, administration of short-acting granulocyte colony-stimulating factor; camera, clinical imaging (CT and PET-CT, see Figure 2). The four time points with targeted NGS of cfDNA are marked with a circled asterisk. (**B**) Levels of serum lactate dehydrogenase (LDH). The upper limit of normal is 244 U/L. (**C**) White blood cell count (leukocyte count).

## Data Availability

The data presented in this study are available on request from the corresponding author. The data are not publicly available due to patient privacy protection.

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
