# Peer review of "Circulating Tumor DNA Profiling of a Diffuse Large B Cell Lymphoma Patient with Secondary Acute Myeloid Leukemia"

_cancers, 2022, doi:10.3390/cancers14061371_

Round 1
Reviewer 1 Report
In this case report, the Authors perform NGS profiling of ctDNA in conjunction with ddPCR quantitation from the patient's blood serum to define the landscape of potential diagnostic driver mutations in DLBCL and therapy-related tAML. The major conclusion from their work reinforces the "diagnostic importance of ctDNA in DLBCL as well as the importance of repeated ctDNA sequencing combined with focused digital PCR assays to display the dynamic mutational landscape during the clinical course." . Overall the work is well presented and the manuscript is well and clearly written. The Authors identify objective limitations of their approach (eg. "could not directly compare of ddPCR and NGS due to the consumption of the entire sample", or "plasma collection for ctDNA prior to DLBCL" to help interpret TP53 p.G245S and PDGFRA p.T276M constant detection).
It would be helpful for the readers if they could briefly discuss, on the basis of available literature or their own unpublished observations, in what percentage of DLBCL patients the potential driver TP53 p.R273L, PDGFRB p.T140M mutations could be detected and thus what the strength of their diagnostic approach would be like if using in a diagnostic panel. Also, how common is the secondary , FLT3 p.D839G variant common to therapy-related AML?
Author Response
"Please see the attachment.

Reviewer 2 Report
This case report describes a case of DLBCL which progressed to AML within 6 months of initial diagnosis. I have concerns regarding the primary diagnosis in this case as well as the diagnosis of t-AML. Since no pathological work-up either at baseline or at a follow-up time point (such as cytochemistry, immunophenotyping etc.) are available, it is not reasonable to interpret the case as t-AML. Besides, the rise in leukocyte count was related to G-CSF administration.
The authors may be asked to add the diagnostic workup of the case to substantiate their diagnosis in this case.
Round 2
Reviewer 2 Report
This paper demonstrates the rationale of disease monitoring using ctDNA in an interesting case of DLBCL with t-AML.
The authors have addressed the comments raised in the previous review. However, it would be nice if they could add two figures- 1) composite showing HPE findings along with IHC useful for diagnosing DLBCL in this case and 2) dot plots of peripheral blood sample demonstrating FCMI data at the time of diagnosis of AML
Author Response
We thank the reviewer for his/her comments. We have added a new Figure 1. Figure 1A shows the HPE findings along with IHC from lymph node biopsies for diagnosing DLBCL. Figure 1B shows cytomorphology and immunophenotyping (dot plots) from peripheral blood for diagnosing AML.